# A Turn-On Fluorescence Sensor Based on Nitrogen-Doped Carbon Dots and Cu^2+^ for Sensitively and Selectively Sensing Glyphosate

**DOI:** 10.3390/foods12132487

**Published:** 2023-06-26

**Authors:** Ziqiang Li, Shuang Liang, Li Zhou, Fengjian Luo, Zhengyun Lou, Zongmao Chen, Xinzhong Zhang, Mei Yang

**Affiliations:** 1Tea Research Institute, Chinese Academy of Agricultural Sciences, Hangzhou 310008, China; 2Key Laboratory of Biology, Genetics and Breeding of Special Economic Animals and Plants, Ministry of Agriculture and Rural Affairs, Hangzhou 310008, China; 3College of Plant Protection, Jilin Agricultural University, Changchun 130118, China

**Keywords:** carbon quantum dots, fluorescence, glyphosate detection, water, rice

## Abstract

Glyphosate has excellent herbicidal activity, and its extensive use may induce residue in the environment and enter into humans living through the food chain, causing negative impact. Here, water-soluble 1.55 nm size nitrogen-doped carbon quantum dots (NCDs) with strong blue fluorescence were synthesized using sodium citrate and adenine. The maximum excitation and emission wavelengths of NCDs were 380 nm and 440 nm, respectively. The above synthesized NCDs were first used for the construction of a fluorescence sensor for glyphosate detection. It was found that Cu^2+^ could quench the fluorescence of NCDs effectively through the photoinduced electron transfer (PET) process, which was confirmed using fluorescence lifetime measurements. Additionally, the fluorescence was restored with the addition of glyphosate. Hence, a sensitive turn-on fluorescence sensor based on NCDs/Cu^2+^ for glyphosate analysis was developed. The LODs of glyphosate for water and rice samples were recorded as 0.021 μg/mL and 0.049 μg/mL, respectively. The sensor was applied successfully for ultrasensitive and selective detection of glyphosate in environmental water and rice samples with satisfied recoveries from 82.1% to 113.0% using a simple sample pretreatment technique. The proposed strategy can provide a significant potential for monitoring glyphosate residue in water and agricultural product samples.

## 1. Introduction

Glyphosate [N-(phosphonomethyl) glycine] is a broad-spectrum organophosphorus herbicide which was developed by the Monsanto Company in the 1970s, and is commonly used to control annual weeds in vegetables and tea, etc., because of its superior herbicidal activity [1]. The market demand for glyphosate is large, and the annual output of glyphosate amounts to 825,800 tons around the world [2]. However, it can cause glyphosate residue pollution in the water, soil and plants because of its excessive spraying and high water solubility. For example, studies have found that glyphosate residues in soybeans and California highway surfaces are as high as 17 mg/kg and 10 mg/L, respectively [3,4]. This can eventually enter human bodies through the food chain [5]. In addition, glyphosate has been classified as “probably carcinogenic to humans (Group 2A)” by the international agency for research on cancer (IARC) [6]. Some organizations including the European Commission (EC) and the United States Environmental Protection Agency (EPA) have established correspondent maximum residual levels (MRLs) of glyphosate [7]. For instance, the EPA and EC set MRLs for glyphosate in water at 0.7 mg/L [8] and in most plants at 0.1 mg/kg [9], respectively. Consequently, it is necessary to develop effective and accurate technologies to determine glyphosate residues.

The high polarity and absence of chromophores and fluorophores in glyphosate make its detection challenging [10]. At present, except for the conventional analysis methods for glyphosate such as chromatography and/or mass spectrometry [11,12], capillary electrophoresis (CE) [13] and ion chromatography (IC) [14] techniques are also included. Although the above methods are highly sensitive and can provide reliable results, there are some ineluctable drawbacks, such as complex derivatization and time-consuming steps, requirements of expensive bulky instruments and professional skills. In recent years, some novel methods, including surface-enhanced Raman scattering spectroscopy (SERS spectroscopy) [15], enzyme-linked immunosorbent assay (ELISA) [16], biosensor [17] and fluorescence sensor [18] have been developed for the analysis of glyphosate. These methods provide advanced detection techniques for glyphosate and overcome some weaknesses of the traditional methods, such as requiring a long time and high cost, etc. In recent years, fluorescence sensors based on various materials like organic dyes, quantum dots, metal-organic frameworks and fluorescent proteins have received significant interest because of their excellent detecting capabilities [19]. Quantum dots (QDs) are a type of new fluorescent nanoparticles with a size of 1–10 nm. Compared with traditional fluorescent dyes, QDs have the characteristics of high fluorescence quantum yield, stable luminescence, wide excitation spectrum, narrow emission spectrum and tunable fluorescence, etc., and have become one of the most frequently used detection materials in fluorescent sensors [20]. For example, cadmium telluride (CdTe) QDs have been used to structure a ratiometric fluorescence nanoprobe for the detection of carbaryl in apples through changing the color tonality [21]. Bala et al. [22] constructed a “turn-off” mode fluorescence sensor based on CdTe@CdS QDs, cationic polymers (PGPMA) and aptamers for the detection of malathion. Nevertheless, the synthesis of traditional CdTe metal quantum dots needs complex synthesis steps and highly toxic raw materials, which limit their practical application in detection [23]. Recently, carbon dots (CDs) with low toxicity, high water solubility, good biocompatibility, chemical inertness, photo-stability and simple synthesis steps have become an even better alternative compared with traditional CdTe quantum dots [24]. Up to now, many fluorescence sensors based on CDs have been applied to detect different kinds of substances, such as pesticides [25], heavy metal ions [26,27], other environmental pollutants [28], medicine [29] and protein [30]. Functionalization, including surface passivation and heteroatom doping, is an efficient approach to improve the fluorescence performance of CDs. Non-metal atom doping CDs doped with N, P, S, B and other elements have attracted attention because of their simple synthesis and high quantum yield, especially nitrogen-doped CDs (NCDs) [31]. Therefore, NCDs have a good prospect for the construction of food safety analysis sensors as excellent fluorescent nanomaterials.

Herein, a turn-on fluorescence sensor based on NCDs applied to glyphosate detection was constructed. The NCDs were synthesized in this study using sodium citrate and adenine as a carbon source and nitrogen source, respectively. According to the principle in Figure 1, the fluorescence intensity of NCDs could be quenched by Cu^2+^ through a photoinduced electron transfer (PET) process, and the assay for glyphosate detection was achieved through the competitive combination of Cu^2+^ between glyphosate and NCDs, resulting in fluorescence restoring of the sensor. Additionally, an as-constructed NCD/Cu^2+^ sensor was applied for the determination of glyphosate in environmental water and rice samples, providing an effective analysis strategy for organic phosphorus pesticide detection in the field of environmental safety.

## 2. Materials and Methods

### 2.1. Reagents and Materials

The sodium citrate (98%), adenine (99.5%), sodium hydroxide (96%), hydrochloric acid (37%), CuSO_4_·5H_2_O (98%) and MnCl_2_·4H_2_O (99%), CaCl_2_ (96%), ZnCl_2_ (98%), AlCl_3_·6H_2_O (97%), KCl (99%), MgCl_2_ (99%), NaCl (99.5%), FeCl_3_·6H_2_O (98%), FeCl_2_·4H_2_O (99%), CdCl_2_·2.5H_2_O (98%), CrCl_3_·6H_2_O (98%), AgCl (99.5%) and PbCl_2_ (99.99%) were obtained from Aladdin Reagent Company (Shanghai, China). Glyphosate (1000 mg/L), glufosinate-ammonium (98.6%), difenoconazole (1000 mg/L), imidacloprid (99.8%), acetamiprid (99.9%), methamidophos (1000 mg/L), methomyl (99.6%), thiamethoxam (99.9%), buprofezin (99.1%), nitenpyram (99.4%), indoxacarb (1000 mg/L) and tolfenpyrad (1000 mg/L) were purchased from ANPEL Laboratory Technologies (Shanghai, China). The dialysis bag was purchased from Spectrum Laboratories Inc. (MWCO = 1000, Rancho Dominguez, CA, USA). The PBS buffer solutions (pH = 7.4, 0.1 M) were obtained from Wokai Biotechnology Co., Ltd. (Beijing, China). The purified water used in the whole work was obtained from Hangzhou Wahaha Group Co., Ltd. (Hangzhou, China), without excess processing. The buffer solutions used in the experiments were diluted to 0.01 M with purified water, and the pH was adjusted with NaOH (1 M) and HCl (1 M).

### 2.2. Synthesis of NCDs

The NCDs were synthesized based on a hydrothermal procedure reported previously with slight modifications [32]. Briefly, sodium citrate (0.0774 g) and adenine (0.1013 g) were added to a 100 mL glass beaker with 20 mL purified water. After ultrasonic vibration for 5 min, the homogeneous mixed solution was transferred into a Teflon-lined autoclave and further heated at 200 °C for 4 h using an oven. Subsequently, the solution was cooled to room temperature. The obtained light-yellow product was preliminarily purified through a 0.22 μm filter membrane and further purified with a 1000 Da dialysis bag for 48 h. Finally, the NCDs solution was diluted 10 times with purified water and stored away from light in a 4 °C refrigerator for subsequent experiments.

### 2.3. Characterization and Photophysical Investigation

The fluorescence spectra of NCDs were recorded using an F-4700 fluorescence spectrophotometer (Hitachi, Japan). The measurement of UV-Vis absorbance spectrum of NCDs was performed using a Varioskan LUX multimode reader (Thermo Fisher Scientific, Waltham, MA, USA). Transmission electron microscopy (TEM) test was carried out via an FEI Tecnai G2 F20. Fourier transform infrared (FT-IR) spectrum was used to investigate the surface groups of NCDs using a Nicolet iS20 Fourier Transform Infrared Spectrometer (Thermo Fisher Scientific, USA). The synthesized CD solution was diluted to a certain multiple before TEM and FT-IR analysis. An X-ray Photoelectron Spectroscopy (XPS) spectrometer (Thermo Scientific Nexsa, Waltham, MA, USA) was utilized to obtain the XPS spectra. The freeze-dried CD powder was quickly transferred to a vacuum glove box before XPS test. A steady/transient fluorescence spectrometer (Edinburgh FLS1000, Edinburgh, UK) was used for the fluorescence lifetime analysis.

### 2.4. Sample Preparation for Glyphosate Analysis

Rice and environmental water samples (tap water and river water) were collected from the local market, our laboratory and Song River (Hangzhou, China), respectively. The rice samples were ground into powders. The rice powder sample (5.00 g) was dispersed in 50 mL purified water, sonicated for 15 min and centrifuged at 10,000 rpm for 10 min. Then, the rice supernatant and the collecting water samples were filtered with 0.22 μm filters to remove water-insoluble organics and particle impurities before detection. The pretreated rice and environmental water samples were spiked with different concentrations of glyphosate standard solutions and analyzed according to the following steps. Generally, a stock NCD solution of 100 μL was added to a 2 mL centrifuge tube with 700 μL PBS buffer solution (pH = 10, 0.01 M). Then, 100 μL Cu^2+^ solution (25 μM CuSO_4_) was immediately added and reacted for 1 min at ambient temperature. Afterwards, 100 μL of the spiked samples was introduced into the above mixture. After thoroughly vortexing and incubating for 3 min, the fluorescence spectroscopy measurement was carried out at 380 nm excitation wavelength using a fluorescence spectrophotometer.

## 3. Results and Discussion

### 3.1. Characterization of NCDs

The NCDs were characterized using TEM, XPS and FT-IR to determinate the morphology and surface structure. The TEM images in Figure 1a show that the NCDs presented spherical morphology and dispersed uniformly, indicating excellent water solubility. Nano Measurer 1.2 software was employed to calculate NCD particle diameter distribution by randomly selecting 150 particles. The histogram in Figure 1b displays that NCDs were distributed in a narrow range from 0.7 to 2.5 nm with an average particle size of 1.55 nm.

FT-IR spectrum was used for investigating surface functional groups and chemical bonds of NCDs. The FT-IR spectrum of NCDs appearing in Figure 1c displayed a broad band at 3394 cm^−1^, which was attributed to the N-H and O-H stretching vibration [33,34]. The strong absorption peak at 1572 cm^−1^ could belong to C=O bending vibration and the asymmetric stretching vibration of the carboxyl anions. The absorption at 1393 cm^−1^ corresponded to C-H bending vibration and the symmetric stretching vibration of the carboxyl anions [35]. The FT-IR analysis results revealed the existence of abundant amino and oxygen functional groups on the carbon frame surface.

The XPS spectra were utilized to further confirm surface element composition and distribution of NCDs. As illustrated in Figure 1d, the full scan XPS spectrum exhibited four distinct characteristic peaks at 285.0, 400.1, 531.0 and 1071.1 eV, corresponding to C1s (51.82%), N1s (5.65%), O1s (30.92%) and Na1s (9.79%), respectively. In detail, the C1s high-resolution spectrum (Figure 1e) showed six peaks at 383.9, 284.8, 285.4, 286.1, 286.6 and 288.3 eV, ascribed to the C=C, C-C, C-N, C-O, C=N and C=O groups, respectively. The three peaks at 399.1, 400.4 and 401.2 eV presented in the N1s high-resolution spectrum (Figure 1f) were assigned to -NH_2_, N-5 (Pyrrolic N) and N-Q (Quaternary N). Additionally, the O1s high-resolution spectrum (Figure 1g) revealed two peaks at 531.2 (C-O) and 532.7 eV (C=O). The above results were consistent with the FTIR spectra.

### 3.2. Optical Properties of NCDs

The photoluminescence properties of prepared NCDs were further studied using UV−Vis absorption spectroscopy, fluorescence spectra and fluorescence stability. The UV−Vis absorption spectrum (yellow line in Figure 2a) exhibited a characteristic absorption peak at 260 nm, attributed to the π–π* electronic transition of the C=C bond [36]. The inset in Figure 2a clearly indicates that the NCD solution revealed a brilliant blue color under irradiation of a 365 nm UV lamp. As indicated in Figure 2b,c, the fluorescence spectra revealed that NCDs demonstrated the maximum fluorescence emission at 440 nm under an excitation of 380 nm. Furthermore, the NCDs displayed an excitation-independent feature, which might be explained by the uniformity of the NCDs’ diameters and surface states [37].

The optical stability of the NCDs was also evaluated under varying conditions, including ionic strength, UV light irradiation and storage time. As observed in Figure 3a, the fluorescence intensity of NCDs was almost unaffected at the different NaCl concentrations (0–3.0 M) and constructed ionic strength conditions, implying an excellent salt stability of NCDs. The anti-photobleaching performance experiment was carried out based on the method reported previously [38]. Additionally, in order to facilitate the development of subsequent portable detection instruments, the 365 nm UV lamp regarded as the most readily available light was chosen for this continuous ultraviolet lamp irradiation experiment. No obvious fluorescence change was discovered after successive exposure of UV light for 3 h (Figure 3b), confirming the admirable photobleaching resistance property. Finally, the fluorescence intensity of NCDs changed less than 5.5% with storage in darkness for 30 days at 4 °C (Figure 3c). Overall, the above features suggested that the synthesized NCDs had excellent luminescence stability and deserved to be applied to the analysis field.

### 3.3. Optimizing Parameters for Glyphosate Analysis

To achieve the optimal detection conditions for glyphosate, the key factors affecting the optical properties of NCDs, namely pH, the concentrations of Cu^2+^ and incubation time on glyphosate detection were independently explored. Figure 4a demonstrates that along with the increase in the pH of PBS buffer solution in the range of 3 to 11, the fluorescence intensity of NCDs continuously enhanced. However, the fluorescence intensity of NCDs/Cu^2+^ showed a decreasing trend and reached the minimal value at pH 10, suggesting that the fluorescence quenching reaction of NCDs reached the maximum. The results were attributed to the fact that the surface functionalized groups of NCDs were easily protonated or deprotonated in acid and alkaline medium [39]. In addition, owing to the partial hydrolysis of copper ion in strong alkaline medium, the coordination effect was inhibited, which weakened the fluorescence quenching reaction [23]. As the above results show, Cu^2+^ could hardly quench the fluorescence of NCDs at a pH of less than six, and the fluorescence of the sensor was not significantly restored with the addition of glyphosate. Moreover, when the pH was at 11, the glyphosate and Cu^2+^ in a strong alkaline solution were both unstable. Thereafter, the optimal pH for glyphosate detection was further validated through fluorescence-restoring experiments with different pH values from 6 to 10. As depicted in Figure 4b, the maximum fluorescence-restoring value (F_2_−F_1_: the fluorescence intensity difference of the NCDs/Cu^2+^ system in the presence and absence of glyphosate) was obtained at pH 10. All the above results indicated that the change in pH seriously affected the sensitivity of the proposed sensor, which was similar to the results of previous studies [19]. Therefore, a pH of 10 was selected as the optimum pH for subsequent experiments.

Additionally, it was also vital to choose an appropriate Cu^2+^ concentration, which was the key factor deeply affecting the quenching efficiency of NCDs. The impact of varied Cu^2+^ concentration on the fluorescence quenching was examined in a concentration range from 5 to 500 μM. A good linear relationship, shown in Figure 5a, was observed between Cu^2+^ concentration and fluorescence quenching intensity (F_0_ − F_1_: the fluorescence intensity difference of the NCD system in the presence and absence of Cu^2+^) when the Cu^2+^ concentration was in the range of 5 to 25 μM. Additionally, the fluorescence quenching gradually stabilized with Cu^2+^ concentrations more than 25 μM. Thus, 25 μM Cu^2+^ was selected for this research.

The effect of response time on the glyphosate detection was also explored. Figure 5b found that, by adding a certain amount of glyphosate, the fluorescence-restoring property of Cu^2+^-quenched NCDs immediately reached a maximum value within 3 min and remained stable afterwards. Such rapid reaction time indicated the fast-response characteristics of the proposed NCDs/Cu^2+^ sensor platform for glyphosate detection. Hence, 3 min was determined to be the optimal response time.

### 3.4. Selectivity and Sensitivity of Glyphosate Detection

In order to appraise the feasibility of the proposed approach, the effects of other potential metal ions (Mn^2+^, Ca^2+^, Zn^2+^, Al^3+^, K^+^, Mg^2+^, Na^+^, Fe^3+^, Fe^2+^, Cr^3+^, Cd^2+^, Ag^+^ and Pb^2+^) on the fluorescence quenching were first verified under the optimal conditions. As shown in Figure 6a, only with the introduction of Cu^2+^ can the fluorescence of NCDs be dramatically quenched, while the presence of other metal ions induced a negligible fluorescence quenching effect. Such behaviors were mainly ascribed to the higher thermodynamic affinity and faster chelation process of Cu^2+^ and some groups on the surface of NCDs [40], implying that the as-prepared NCDs exhibited high selectivity for Cu^2+^ compared with other common metal ions.

Furthermore, the selectivity of the constructed NCDs/Cu^2+^ sensor towards glyphosate was explored by contrasting the fluorescence-restoring ability in the presence of various pesticides, including glufosinate-ammonium, difenoconazole, imidacloprid, acetamiprid, methamidophos, methomyl, thiamethoxam, buprofezin, nitenpyram, indoxacarb and tolfenpyrad. The results in Figure 6b illustrate that glyphosate considerably enhanced the fluorescence intensity of NCDs/Cu^2+^. The addition of glufosinate-ammonium caused slight fluorescence restoring of the NCDs/Cu^2+^ system because of the structural similarity between glufosinate-ammonium and glyphosate. However, it did not affect the detection of glyphosate. This unapparent phenomenon was regarded as the greater distance between the amino and phosphonate groups of glufosinate-ammonium, which made it difficult to form a steric chelate between glufosinate-ammonium and Cu^2+^. Moreover, the other mentioned pesticides lacked certain groups that could coordinate with Cu^2+^, which was hard to cause fluorescence restoring [41].

In summary, it was clear that our designed fluorescence sensor displayed high selectivity for detecting glyphosate. Accordingly, to estimate the sensitivity of glyphosate determination, the response of fluorescence emission spectra in the presence of various concentrations of glyphosate was recorded at λ_em_ = 440 nm after exciting at 380 nm. The calibration curves of glyphosate in water (Figure 6c) and rice samples (Figure 6d) were obtained and demonstrated that the fluorescence intensity of NCDs/Cu^2+^ gradually elevated with the increase in glyphosate concentration. Furthermore, the best linear fits for the glyphosate concentration in water and rice were obtained (Table 1). The linear ranges showing good linear relationships for the concentration of glyphosate were 0.10–10 μg/mL in water and 0.10–8.0 μg/mL in rice, respectively. The slight linear range difference between pure water and rice samples could be attributed to the matrix complexity. According to the calculation method (3σ/S, where σ is the standard deviation of the blank NCDs sample (*n* = 10) and S is the slope of the calibration curve), the limits of detection (LODs) toward glyphosate were estimated to be 0.021 μg/mL in water and 0.049 μg/mL in rice, respectively. In addition to that, it was apparent that our designed sensor presented a broader linear response range and comparable LODs compared with most of the other sensors for measuring glyphosate in recent years (Table 2), and the lower LODs met the demand of the MRL of glyphosate in water and rice, proving the sensor’s good application for the determination of glyphosate. Compared with the traditional chromatographic method of glyphosate detection, the constructed sensor in this study provided a low-cost, rapid and accurate method. However, the sensitivity of the conventional method is still higher than that of the novel constructed fluorescence sensor in this study due to the limitation of the fluorescence performance of NCDs. Compared with the synthesized single nitrogen-doped CDs, multi-element co-doping can significantly improve the fluorescence performance of CDs [31]. Our next study will focus on the functionalization of CDs to improve the detection performance of the sensor, which may provide a reference idea for the development of advanced glyphosate rapid detection technology.

### 3.5. Fluorescent Sensing Principle of Glyphosate

The fluorescence intensity of NCDs can be significantly decreased in the presence of Cu^2+^, as shown in Figure 7a. The quenched fluorescence could be restored to a certain extent with the addition of glyphosate. However, when only NCDs and glyphosate were present in the system, the fluorescence intensity of NCDs essentially did not change, and the black dotted line overlapped with the green line in Figure 7a, indicating that no reaction occurred between glyphosate and NCDs. To further clarify the quenching mechanism between Cu^2+^ and NCDs, the fluorescence lifetimes of NCDs and NCDs/Cu^2+^ were measured. As depicted in Figure 7b, the average fluorescence lifetime of NCDs/Cu^2+^ was 4.22 ns, which was different from that of NCDs (3.22 ns). The results suggested that the fluorescence of NCDs was decreased through the occurrence of the PET process from NCDs to Cu^2+^, where the chelates were produced by the coordination between the surface groups of NCDs and Cu^2+^ [20].

### 3.6. Sample Analysis

To investigate the application potential of the proposed sensor in sample analysis, spiked experiments of tap water, river water and rice were carried out. Different spiked concentration samples (0, 1.0, 3.0, 5.0 μg/mL) were detected using the constructed sensor after a simple filtering procedure. Then, the concentration and recovery of glyphosate were calculated according to the established calibration curves of the water and rice matrix. As summarized in Table 3, the recoveries of tap water, river water and rice were 83.8–105.9%, 85.4–113.0% and 82.1–111.1%, respectively, and all the relative deviation standards (RSDs) were lower than 15%. Recovery and RSD results met the requirements of pesticide residue analysis. The above results revealed that the proposed NCDs/Cu^2+^ sensor was suitable for the analysis of glyphosate in environmental water and rice samples.

## 4. Conclusions

The NCDs that were synthesized using sodium citrate and adenine as a carbon source and nitrogen source, respectively, were first successfully used for the construction of a turn-on fluorescence sensor for glyphosate detection in this work. Additionally, the prepared NCDs showed good salt tolerance, anti-photobleaching and storage stability. The fluorescence of NCDs could be specifically quenched by Cu^2+^ through a PET process, and the fluorescence of NCDs was restored due to the stronger chelation between glyphosate and Cu^2+^. The as-constructed fluorescence sensor presented a preeminent selectivity and analytical property. The concentration of glyphosate in the water and rice matrix showed a good linear relationship in the range of 0.10–10 μg/mL and in the range of 0.10–8.0 μg/mL, respectively. The LODs of glyphosate were 0.021 μg/mL in water and 0.049 μg/mL in rice, respectively. Compared with most of the other sensors for measuring glyphosate in recent years, the designed sensor presented a broader linear response range and comparable LODs. Finally, the potential of the constructed sensor for application in real samples was verified by spiked recovery experiments at three concentration levels (1.0, 3.0, 5.0 μg/mL). The recoveries of tap water, river water and rice were 82.1–113.0%, respectively. The results indicated that the NCDs/Cu^2+^ fluorescence sensor presented potential applications in the determination of glyphosate in environmental water and rice samples, which could also offer a novel tool for substance detection in the field of environment safety. However, the constructed sensor has some gaps in sensitivity and the application in a complex matrix compared with traditional methods. More efforts are needed to improve the fluorescence performance of NCDs and develop methods for glyphosate detection in complex substrates in the future.

## Data Availability

All related data and methods are presented in this paper. Additional inquiries should be addressed to the corresponding author.

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
