# Peer review of "A Turn-On Fluorescence Sensor Based on Nitrogen-Doped Carbon Dots and Cu2+ for Sensitively and Selectively Sensing Glyphosate"

_foods, 2023, doi:10.3390/foods12132487_

Round 1

Reviewer 1 Report

In this paper, Li et al described “A turn-on fluorescence sensor based on nitrogen-doped carbon dots and Cu2+ for sensitively and selectively sensing glyphosate”.

Comments:

1.      The quality of figures is not enough and should be improved.

2.      Page 2, Line 70, add this reference for heavy metal ions: https://doi.org/10.1016/j.saa.2023.122448

3.      In Fig. 7a, green line of NCDs is not distinguishable. It is mentioned that the green line is overlapped with yellow one, but It should be shown with a dashed line.

Reviewer 2 Report

The paper presents a fluorescence sensor for the detection of glyphosate in water and rice samples. The article is, in general, well written. I suggest the following revisions:

1) The figures are in general too small and barely readable. Please, improve the quality of the figures and increase the text size since it is almost not readable when paper printed.

2) Fig. 3(b) shows the effects of continuous UV radiation at 365nm on fluorescence intensity of NCDs. How the peak wavelength of 365nm has been chosen? Why not using the wavelength of 380nm used in the measurements?

3) In section 3.3 the effect of pH of PBS buffer is investigated in the range 3 – 11. However, in Fig. 4 (b) the fluorescence restoring value is investigated in the subrange 6 – 10. Why not using the full range 3 – 11?

4) In Table 2 the performance of the proposed sensor is compared to other sensors from literature. I think more discussion on how the proposed sensor improves the state-of-the-art can be useful. For example, ref 37 presents a comparable linear range and better LOD.

5) The authors should fix some errors and typos. For example: at the beginning of the abstract “enter” and not “inter”; at page 2 line 62 CdTe should be written extensively the first time it is used.

Reviewer 3 Report

Authors present a proof-of-concept for detection of glyphosate in water and rice. They used the water-soluble 1.55 nm size nitrogen-doped carbon quantum dots (NCDs). The proposed sensor was applied successfully for ultrasensitive and selective detection of glyphosate in environmental water and rice samples. The presented idea and results are interesting thus the manuscript can be accepted after minor revision.

Scheme 1 – should be changed in terms of readability (remove dark backbground). In my opinion Authors should re-design the whole scheme to make it more clear for the reader. Please move the Scheme 1 to materials and Methods.

Line 100 – what does "pure water" mean? Please add some details about purification method.

Please increase the size of the fonts in Figure 1-7.

Please add information about methods of sample preparation for TEM, FTIR and XPS.

Figure 6d – both axes should start with zero in the same spot on axis (see Figure 6c).

Authors should elaborate more on LOD and recovery in terms of comparing their method with alternative methods. Is it better, worse, comparable?

I have no remarks about English language used in the manuscript.

Round 2

Reviewer 2 Report

The authors have revised the paper according to the Reviewer comments. It is now suitable for publication.
